# Discrete-Time Quantum Walk on Multilayer Networks

**DOI:** 10.3390/e25121610

**Published:** 2023-11-30

**Authors:** Mahesh N. Jayakody, Priodyuti Pradhan, Dana Ben Porath, Eliahu Cohen

**Affiliations:** 1Faculty of Engineering and the Institute of Nanotechnology and Advanced Materials, Bar-Ilan University, Ramat Gan 5290002, Israel; dana9494@gmail.com (D.B.P.); eliahu.cohen@biu.ac.il (E.C.); 2networks.ai Lab, Department of Computer Science and Engineering, Indian Institute of Information Technology Raichur, Raichur 584135, Karnataka, India; priodyutipradhan@gmail.com

**Keywords:** discrete-time quantum walks, multilayer network, decoherence

## Abstract

A Multilayer network is a potent platform that paves the way for the study of the interactions among entities in various networks with multiple types of relationships. This study explores the dynamics of discrete-time quantum walks on a multilayer network. We derive a recurrence formula for the coefficients of the wave function of a quantum walker on an undirected graph with a finite number of nodes. By extending this formula to include extra layers, we develop a simulation model to describe the time evolution of the quantum walker on a multilayer network. The time-averaged probability and the return probability of the quantum walker are studied with Fourier, and Grover walks on multilayer networks. Furthermore, we analyze the impact of decoherence on quantum transport, shedding light on how environmental interactions may impact the behavior of quantum walkers on multilayer network structures.

## 1. Introduction

Quantum walks (QWs) are the quantum analogs of classical random walks (CRWs). Importantly, QWs contribute to the theoretical and applied studies of quantum computing [1] and quantum algorithms [2]. There are two broad classes of QWs known as discrete-time QWs (DTQW) and continuous-time QWs (CTQW), each of which has significant distinctions in their mathematical formalism [3]. A considerable body of work can be found in the literature which explores the dynamics of linear and cyclic QWs in two and higher-dimensional spaces, as well as on specific graphs [3,4].

In the study of complex systems, multilayer networks play a crucial role as a modeling tool [5,6,7]. A multilayer network consists of nodes and edges, yet the edges exist in separate layers representing different forms of interactions. Multilayer networks are used to understand the evolution of ecological systems [8], complex interactions across multiple layers of biological systems [9], public transportation systems [10] and the structure of financial markets [11]. One can find several studies that utilize the framework of CTQWs to investigate transport properties of multilayer dendrimer networks [12], honeycomb networks [13], and important models of scale-free networks [14]. Nonetheless, it is clear that there has been insufficient exploration of the dynamics of DTQWs on multilayer networks. Therefore, we present a comprehensive study of DTQWs on a multilayer network to address this gap. Several formulations of the DTQWs on specific network structures can be found in the literature. However, defining a DTQWs on an arbitrary network is more difficult than that of a CTQWs [15]. Ref. [16] has proposed a framework for defining QWs on regular graphs. Sometimes, this framework is termed the Shunt-Decomposition model [17]. In [2], it is suggested to add one or more self-loops to each vertex with the purpose of obtaining a regular graph when modeling the QWs on irregular graphs. Redefining the action of the conditional swap operator in [18], Kendon [19] has presented a method to simulate DTQWs on general undirected graphs. Sometimes, this method is identified as the arc-reversal model [17]. Ref. [20] has proposed another framework for QWs on a graph in which the motion of the quantum walker takes place on the edges of the graph rather than the vertices.

In our work, we give a block matrix representation for the state of the quantum walker instead of the usual column vector representation. Then, we derive formulae that imitate the coin toss and the shifting operation of the DTQW. Using these formulae and the block matrix representation, we develop a simulation to mimic the progression of DTQWs on an undirected graph. Later, we extend our framework to mimic DTQWs on multilayer networks.

The paper is organized in the following way. In Section 2, we present our mathematical model for DTQWs on a graph. Section 3 is dedicated to extending our mathematical model to include DTQWs on multilayer networks. Moreover, for comparison purposes, we also model a CRW on a multilayer network in Section 4. Numerical implementation of DTQWs on a toy model and some synthetic multilayer networks are given in Section 5 and Section 6, respectively, along with a detailed analysis of time-average probability, return probability, and decoherence.

## 2. QWs on a Graph

Consider a finite undirected graph G={V,E} where $V=\{v_{1},\ldots ,v_{n}\}$ is the set of vertices (nodes) and $E=\{\left(v_{i},v_{j}\right)\;|\;v_{i},v_{j}\in V\}$ is the set of edges (connections). Note that, the graphs we studied here are finite as opposed to the unrestricted line or the integer line we used to define the QWs on a line [21]. Moreover, our study considers only simple graphs (i.e., graphs without self-loops or parallel edges). We denote the adjacency matrix corresponding to G as $A\in R^{n\times n}$ which is defined in the following way
(1)$$a_{ij}=\left\{\begin{array}{cc} 1 & \;\mathrm{If}\;(v_{i},v_{j})\in E\\ 0 & \;\mathrm{Otherwise} \end{array}\right.$$
The positive integers n=|V| and m=|E| represent the number of nodes and edges in G, respectively. The number of edges linked to a particular node $v_{i}$ is referred to as its degree and denoted by $d_{i}=\sum_{j=1}^{n}a_{ij}$.

Let us now model the propagation of a QW on G. For convenience, let us relabel the vertices of G or, in the QWs’ terminology, the position states of the QW as ${|x\rangle }_{p}$ where x=1,…,n. Then, the set of vertices $V={\{|1\rangle }_{p}{,|2\rangle }_{p}{,\ldots ,|n\rangle }_{p}\}$ becomes the position basis set of the QW which spans the position of the Hilbert space $H_{p}$. Let us denote the subspace spanned by the basis element ${|x\rangle }_{p}$ in *V* as $H_{p}^{\left(x\right)}$. That is, $H_{p}^{\left(x\right)}=\mathrm{span}{\{|x\rangle }_{p}\}$ where ${|x\rangle }_{p}\in V$. Now, for each vertex ${|x\rangle }_{p}$, let us assign a coin Hilbert space $H_{c}^{\left(x\right)}$ spanned by the coin basis ${\{|r\rangle }_{c}\;\left|\;r=1,\cdots ,d_{x}\right\}$ where $d_{x}$ is the degree of the vertex ${|x\rangle }_{p}$. Note that, the dimension of the coin Hilbert space $H_{c}^{\left(x\right)}$ is $d_{x}$. To gain an advantage in simulation, we adopt the following strategy to modify the labeling of the coin states. We define a set $B_{x}$ that comprises the labels of the vertices adjacent to the vertex ${|x\rangle }_{p}$ as $B_{x}={\{y\;:\;\mathrm{vertex}\;|x\rangle }_{p}{\;\mathrm{and}\;|y\rangle }_{p}\;\mathrm{have}\;a\;\mathrm{connection}\}$ where $|B_{x}|=d_{x}$. Now, we define a function as, $f_{x}\left(r\right)=r\mathrm{th}\;\mathrm{element}\;\mathrm{of}\;B_{x}$. Since the parallel edges are excluded in our study, the function $f_{x}\left(r\right)$ is a bijection. Without the loss of generality, we always arrange the elements of $B_{x}$ in an ascending order. Now, we can denote the basis of the coin Hilbert space $H_{c}^{\left(x\right)}$ as ${\{|f_{x}\left(r\right)\rangle }_{c}\;\left|\;r=1,\cdots ,d_{x}\right\}$. Using $f_{x}\left(r\right)$, we have labeled the coin states of each vertex in terms of the edges connected to it. Such an approach can be found in [18]. Now, the state vector of the quantum walker at position ${|x\rangle }_{p}$ at time *t* can be written as
(2)$$|\psi \left(x,t\right)\rangle =\sum_{r=1}^{d_{x}}\alpha_{x,f_{x}\left(r\right)}{\left(t\right)|x\rangle }_{p}{|f_{x}\left(r\right)\rangle }_{c}$$
where $\alpha_{x,f_{x}\left(r\right)}\left(t\right)\in C$ are called the probability amplitudes, ${|x\rangle }_{p}\in H_{p}^{\left(x\right)}$, $|f_{x}{\left(r\right)\rangle }_{c}\in H_{c}^{\left(x\right)}$ and $|\psi \left(x,t\right)\rangle \in H_{p}^{\left(x\right)}⊗H_{c}^{\left(x\right)}$. Here, $dim(H_{p}^{\left(x\right)})=1$, $dim\left(H_{c}^{\left(x\right)}\right)=d_{x}$ and $dim\left(H_{p}^{\left(x\right)}⊗H_{c}^{\left(x\right)}\right)=d_{x}$. For each position *x*, the coefficient $\alpha_{x,r}\left(t\right)=0$ whenever $r\notin B_{x}$. Our next task is to write an expression for the total wave function of the quantum walker on G at time *t*. For that, we need to sum the state vectors |ψ(x,t)〉 in (Equation 2) over all the vertices. However, we cannot perform such a summation because the state vectors |ψ(x,t)〉, corresponding to each vertex, reside in different composite Hilbert spaces as the size of $H_{c}^{\left(x\right)}$ changes with the degree of the node *x*. Therefore, to perform such a summation, one needs to combine the set of composite Hilbert spaces ${\left\{H_{p}^{\left(x\right)}⊗H_{c}^{\left(x\right)}\right\}}_{x}$ in a reasonable manner to form a bigger Hilbert space that includes all the state vectors. The operation of the direct sum of the vector spaces paves a way to combine the composite Hilbert spaces to cater to our demand. Let us define $H=\overset{n}{\underset{x=1}{⨁}}\left(H_{p}^{\left(x\right)}⊗H_{c}^{\left(x\right)}\right)$ where ⊕ denotes the external direct sum of Hilbert spaces and $\dim\left(H\right)=\sum_{x=1}^{n}d_{x}$. According to the definition of the global Hilbert space *H*, it is obvious that for each vertex *x*, the state vector |ψ(x,t)〉∈H. Hence, the total wave function of the quantum walker at time *t* can be calculated by summing |ψ(x,t)〉 in (Equation 2) over all the vertices. Then, the total wave function at time *t* can be written as
(3)$$|\psi_{t}\rangle =\sum_{x=1}^{n}\sum_{r=1}^{d_{x}}\alpha_{x,f_{x}\left(r\right)}{\left(t\right)|x\rangle }_{p}{|f_{x}\left(r\right)\rangle }_{c}$$
where $|\psi_{t}\rangle \in H$ and $\sum_{x=1}^{n}\sum_{r=1}^{d_{x}}{\left|\alpha_{x,f_{x}\left(r\right)}\left(t\right)\right|}^{2}=1$. Coin operator $C^{\left(x\right)}$, which acts on the coin states associated with the vertex ${|x\rangle }_{p}$, holds the transition probabilities from ${|x\rangle }_{p}$ to its neighboring vertices. Hence, $C^{\left(x\right)}$ can be defined as
(4)$$C^{\left(x\right)}=\sum_{i=1}^{d_{x}}\sum_{j=1}^{d_{x}}C_{ij}^{\left(x\right)}|f_{x}\left(i\right)\rangle \langle f_{x}\left(j\right)|$$
where ${\{|f_{x}\left(r\right)\rangle }_{c}{\}}_{r=1}^{d_{x}}$ are the basis elements of $H_{c}^{\left(x\right)}$. The coin coefficients $C_{ij}^{\left(x\right)}\in C$ are chosen in such a way that the condition of unitarity of the QW is preserved, i.e., the total probability is unity at all time steps. Hence, ${\left(C^{\left(x\right)}\right)}^{†}C^{\left(x\right)}=C^{\left(x\right)}{\left(C^{\left(x\right)}\right)}^{†}=I$. By combining each local coin operator $C^{\left(x\right)}$, one can write the global coin operator *C* acting on $|\psi_{t}\rangle \in H$ as follows
(5)$$C=\sum_{x=1}^{n}\sum_{i=1}^{d_{x}}\sum_{j=1}^{d_{x}}C_{ij}^{\left(x\right)}\left|x\rangle \langle x|⊗\right|f_{x}\left(i\right)\rangle \langle f_{x}\left(j\right)|$$
The shift operator of the QW on a graph is defined as follows
(6)$${S|x\rangle }_{p}{|y\rangle }_{c}={|y\rangle }_{p}{|x\rangle }_{c}$$
Note that the abstract forms of *C* and *S* given in (Equation 5) and (Equation 6), respectively, are similar to those used in the arc-reversal model [17]. However, the arc-reversal model and our method significantly differ in numerical simulation (discussed in Appendix A). Both *C* and *S* are unitary operators associated with the Hilbert space *H*. Hence, a single-step progression of the quantum walker on the graph is given by
(7)$$|\psi_{t+1}\rangle =U|\psi_{t}\rangle$$
where U=SC is the evolution operator and *S* and *C* are the shift and coin operators, respectively.

### 2.1. Matrix Representation and Simulation

The local coin operator $C^{\left(x\right)}$, associated with the vertex ${|x\rangle }_{p}$, holds the transition probabilities from ${|x\rangle }_{p}$ to its neighboring nodes. Suppose that the vertex ${|x\rangle }_{p}$ is linked to a $d_{x}$ number of vertices denoted by $|y_{1}\rangle_{p},\cdots ,{|y_{d_{x}}\rangle }_{p}$ and $y_{1}<y_{2}<\ldots <y_{d_{x}}$. Hence, for each $r\in \{1,\cdots ,d_{x}\}$, we can write $y_{r}=f_{x}\left(r\right)$. Then, the block matrix of $C^{\left(x\right)}$ can be written as
(8)$$C^{\left(x\right)}=\begin{array}{cc} & \begin{array}{ccc} \langle f_{x}\left(1\right)| & \cdots & \langle f_{x}\left(d_{x}\right)| \end{array}\\ \begin{array}{c} |f_{x}\left(1\right)\rangle \\ \vdots \\ |f_{x}\left(d_{x}\right)\rangle \end{array} & \left(\begin{array}{ccc} C_{11}^{\left(x\right)} & \cdots & C_{1d_{x}}^{\left(x\right)}\\ \vdots & \vdots & \vdots \\ C_{d_{x}1}^{\left(x\right)} & \cdots & C_{d_{x}d_{x}}^{\left(x\right)} \end{array}\right) \end{array}$$
For example, the local coin operators associated with a four-vertex graph are shown in Appendix A. In the standard mathematical formalism of QWs [21], a column vector represents the state of the quantum walker at time *t*. However, alternatively, one can give a convenient block matrix representation for the total wave function of the quantum walker on a graph given in (Equation 3) as follows
(9)$$|\psi_{t}\rangle =N_{t}=\begin{array}{cc} & \begin{array}{ccc} {|1\rangle }_{c} & \cdots & {|n\rangle }_{c} \end{array}\\ \begin{array}{c} {|1\rangle }_{p}\\ \vdots \\ {|n\rangle }_{p} \end{array} & \left(\begin{array}{ccc} \alpha_{1,1}\left(t\right) & \cdots & \alpha_{1,n}\left(t\right)\\ \vdots & \vdots & \vdots \\ \alpha_{n,1}\left(t\right) & \cdots & \alpha_{n,n}\left(t\right) \end{array}\right) \end{array}$$
In matrix $N_{t}$, the rows represent the position states, and the columns represent the coin states. A single row holds the coefficients corresponding to the coin states associated with a single position. According to the definition of the coefficients of $\alpha_{x,r}\left(t\right)$, some entries of the matrix $N_{t}$ become zero (Appendix A). Such a block matrix representation of the total wave function can be found in the study [22]. One can view the block matrix representation given in (Equation 9) as an adjacency matrix of a weighted graph with time-dependent complex weights. By knowing all the coefficients of $\alpha_{x,r}\left(t\right)$, in other words, all the elements of $N_{t}$, we can uniquely determine the total wave function of the quantum walker on the graph at time *t*. In addition, by appropriately updating the elements of $N_{t}$, one can determine the matrix $N_{t+1}$ and hence the total wave function of the quantum walker at time t+1. The elements of $N_{t}$ are updated in two processes. First, an intermediate matrix ${\tilde{N}}_{t}$ is generated using the following formula
(10)$${\tilde{\alpha }}_{x,f_{x}\left(i\right)}\left(t\right)=\sum_{j=1}^{d_{x}}\alpha_{x,f_{x}\left(j\right)}\left(t\right)C_{ij}^{\left(x\right)}$$
Afterwards, the matrix $N_{t+1}$ is determined by taking the transpose of ${\tilde{N}}_{t}$. The update rule is given by
(11)$$\alpha_{x,f_{x}\left(i\right)}\left(t+1\right)={\tilde{\alpha }}_{f_{x}\left(i\right),x}\left(t\right)$$
Note that the expression in (Equation 10) and the recurrence formula in (Equation 11) correspond to the coin and shift operations of the QW on the graph, respectively. The proof is given in the Appendix B.

### 2.2. Probability Calculation

The probability $P_{q}\left(x,t\right)$ of finding the quantum walker at vertex *x* at time *t* can be calculated using (Equation 2) as follows
(12)$$P_{q}\left(x,t\right)=\sum_{r=1}^{d_{x}}{\left|\alpha_{x,f_{x}\left(r\right)}\left(t\right)\right|}^{2}$$
Note that $P_{q}\left(x,t\right)$ can be determined by summing the elements in the xth row of the matrix $N_{t}⊙N_{t}^{*}$ where ⊙ is the Hadamard product of matrices and $N_{t}^{*}$ is the complex conjugate of the block matrix given in (Equation 9).

## 3. QWs on Multilayer Networks

A multilayer network is a pair M=(G,C) where $G=\{L_{\alpha };\;\alpha \in \left\{1,2,\ldots ,l\right\}\}$ is a family of undirected graphs of $L_{\alpha }=\left\{V_{\alpha },E_{\alpha }\right\}$ (called layers of M) with $V_{\alpha }=\left\{v_{1}^{\alpha },v_{2}^{\alpha },\ldots ,v_{n_{\alpha }}^{\alpha }\right\}$ is the set of vertices and $E_{\alpha }=\left\{e_{1}^{\alpha },e_{2}^{\alpha },\ldots ,e_{r_{\alpha }}^{\alpha }:e_{r_{\alpha }}=\left(v_{i}^{\alpha },v_{j}^{\alpha }\right)\right\}$ is the set of edges in the $L_{\alpha }$ layer of the multilayer network [6]. The positive integers *l*, $n_{\alpha }$ and $r_{\alpha }$ are termed as the number of layers in M, the number of vertices and the edges of the layer $L_{\alpha }$ respectively. Moreover, $C=\{E_{\alpha \beta }\subseteq V_{\alpha }\times V_{\beta }:\;\alpha ,\beta \in \left\{1,2,\ldots ,l\right\},\alpha \neq \beta \}$ is the set of edges between the $L_{\alpha }$ and $L_{\beta }$ layers. The elements of C are crossed layers. Furthermore, the elements of each $E_{\alpha }$ are called the set of intralayer edges, and the elements of each $E_{\alpha \beta }$ (α≠β) are called the interlayer edges of M [23]. Let us denote the adjacency matrices corresponding to each layer $L_{\alpha }$ as $A^{\left(\alpha \right)}=\left(a_{ij}^{\alpha }\right)\in R^{n_{\alpha }\times n_{\alpha }}$ which is defined by
(13)$$a_{ij}^{\alpha }=\left\{\begin{array}{cc} 1 & \;\mathrm{If}\;\left(v_{i}^{\alpha },v_{j}^{\alpha }\right)\in E_{\alpha }\\ 0 & \;\mathrm{Otherwise} \end{array}\right.$$
for $1\leq i,j\leq n_{\alpha }$ and 1≤α≤l where $n_{\alpha }$ is the number of nodes in layer $L_{\alpha }$. The interlayer adjacency matrix corresponding to $E_{\alpha \beta }$ is the matrix $A^{\left[\alpha ,\beta \right]}\in R^{n_{\alpha }\times n_{\beta }}$ given by
(14)$$a_{ij}^{\alpha \beta }=\left\{\begin{array}{cc} 1 & \;\mathrm{If}\;\left(v_{i}^{\alpha },v_{j}^{\beta }\right)\in E_{\alpha \beta }\\ 0 & \;\mathrm{Otherwise} \end{array}\right.$$
By appropriately combining the adjacency matrices corresponding to each layer $L_{\alpha }$ and the interlayer adjacency matrices, one can derive a supra-adjacency matrix [23] which characterizes M. Now, let us define a QW on M. Recall that, in Section 2, we developed a mathematical model to mimic the propagation of a QW on any given undirected graph. According to our model, when the adjacency matrix is given, we define the sets of ${\left\{B_{x}\right\}}_{x=1}^{n}$ along with the set of functions ${\left\{f_{x}\left(r\right)\right\}}_{x=1}^{n}$ and then simulate the evolution of the QW on the graph by updating the elements of the block matrix in (Equation 9). Likewise, one can use the same mathematical model to mimic the propagation of a quantum walker on a multilayer network just by following the same procedure given in Section 2 with the supra-adjacency matrix of M. In a QW, the transition probabilities from vertex *x* to its neighboring vertices are given by the coin operator $C^{\left(x\right)}$. According to the simulation, one can choose suitable coin operators to control the probability flow from one vertex to the neighboring vertices in the multilayer network. When the transition probabilities from a vertex *x* to its neighboring vertices are the same, we say that the QW on the multilayer network is unbiased. To model such an unbiased QW, we can attach a Fourier coin $F^{\left(x\right)}$ to each vertex *x* given by
(15)$$F^{\left(x\right)}=\frac{1}{\sqrt{d_{x}}}\sum_{r=1}^{d_{x}}\sum_{s=1}^{d_{x}}e^{2i\pi \left(r-1\right)\left(s-1\right)/d_{x}}\left|r\rangle \langle s\right|$$
where $d_{x}$ is the degree of vertex *x* [24]. Note that, the relationship of $F^{\left(x\right)}{\left(F^{\left(x\right)}\right)}^{†}={\left(F^{\left(x\right)}\right)}^{†}F^{\left(x\right)}=I$ is preserved by the Fourier coin. In the studies of QWs on graphs, it has been shown that the Grover coin tends to localize the quantum walker around the initial vertex [24]. Hence, it is also worth exploring the Grover’s walk on a multilayer network. The ijth element of the Grover coin $G^{\left(x\right)}$ attached to the vertex *x* on M can be written as
(16)$$G_{ij}^{\left(x\right)}=\left\{\begin{array}{cc} \frac{2-d_{x}}{d_{x}} & \;\mathrm{If}\;i=j\\ \frac{2}{d_{x}} & \;\mathrm{Otherwise} \end{array}\right.$$
where $1\leq i,j\leq d_{x}$ and $d_{x}$ is the degree of vertex *x*. Note that, the Grover coin holds the relationship of $G^{\left(x\right)}{\left(G^{\left(x\right)}\right)}^{†}={\left(G^{\left(x\right)}\right)}^{†}G^{\left(x\right)}=I$. In Section 5 and Section 6, we analyze the dynamics of some QWs on specific multilayer networks with different choices of coin operators. Note that a multilayer network can be conceptualized as a single-layer network with different groups of nodes, each having distinct connection properties. Hence, the general definition of discrete-time QWs on the undirected graph in Section 2 already captures the case of QWs on multilayer networks. However, since the multilayer networks offer a more detailed and comprehensive representation of complex systems [25], the framework for QWs on multilayer networks, which we have given in this paper, can be conveniently used to model and analyze complex scenarios. For instance, our framework for QWs on multilayer networks enables the modeling and analyzing of the situation in which each layer possesses a different coin (Appendix C and Figure A2).

## 4. Classical Random Walk on Multilayer Networks

The propagation of the classical random walk (CRW) on different graph structures is a topic that has been extensively studied [26]. In general, the propagation of a random walker on any given network structure is modeled using the transition probabilities from a vertex *x* to its neighboring vertices [19,27]. By adopting the same concept, one can also define a CRW on a multilayer network [28]. Let $\Omega_{x,y}$ be the transition probability from vertex *x* to *y*. Then, the probability $P_{c}\left(x,t\right)$ of finding the random walker at position *x* at time *t* is given by the following recurrence relations
(17)$$P_{c}\left(x,t\right)=\sum_{r=1}^{d_{x}}\Omega_{f_{x}\left(r\right),x}P_{c}\left(f_{x}\left(r\right),t-1\right)$$
where $d_{x}$ is the degree of the vertex *x* and for each *r*, the function $f_{x}\left(r\right)$ gives the labels of the neighbouring vertices of *x*. When the transition probabilities from a vertex *x* to its neighboring vertices are the same, we say the CRW is unbiased. In usual practice [27], transition probability $\Omega_{x,y}^{\left(ub\right)}$ for the unbiased classical random walk (UBCRW) on any graph structure is defined as follows
(18)$$\Omega_{x,y}^{\left(ub\right)}=\left\{\begin{array}{cc} \frac{1}{d_{x}} & \;\mathrm{If}\;(x,y)\;\mathrm{is}\;\mathrm{connected}\\ 0 & \;\mathrm{Otherwise} \end{array}\right.$$
where $d_{x}$ is the degree of vertex *x*. One can adopt the same definition given in (Equation 18) to model the UBCRW on a multilayer network.

## 5. Numerical Implementation on a Toy Model

In this section, we perform the UBCRW and the Fourier walk (unbiased QW) on a toy multilayer network structure (Figure 1) and examine the flow of probability through various layers. We intend to explore the fundamental differences between classical and quantum dynamics on a multilayer network. From the definition of the multilayer network given in Section 3, one can consider diverse configurations of structures with multilayers. Nonetheless, a two-layer network consisting of two distinct graphs can be understood as the simplest multilayer network. Hence, we consider one such simplest multilayer structure to perform the UBCRW and the Fourier walk. Note that the interlayer edges of the toy model in Figure 1 only link the vertices representing the same entity in different layers. Hence, this network can be classified as a multiplex network, which is a special class of multilayer networks [29]. Let the top and bottom layers be $L_{1}=\left\{V_{1},E_{1}\right\}$ and $L_{2}=\left\{V_{2},E_{2}\right\}$ respectively where $V_{1}=\left\{1,2,3,4\right\}$ and $V_{2}=\left\{5,6,7,8\right\}$ are the set of vertices in each layer and $E_{1}$ and $E_{2}$ are the set of edges corresponding to each layer. The supra-adjacency matrix [23] of the multilayer network in Figure 1 can be written as follows
(19)$$A_{sup}=\left(\begin{array}{cc} A^{\left(1\right)} & I\\ I & A^{\left(2\right)} \end{array}\right)=\left(\begin{array}{cc} \begin{array}{cccc} 0 & 1 & 1 & 1\\ 1 & 0 & 1 & 1\\ 1 & 1 & 0 & 1\\ 1 & 1 & 1 & 0 \end{array} & \begin{array}{cccc} 1 & 0 & 0 & 0\\ 0 & 1 & 0 & 0\\ 0 & 0 & 1 & 0\\ 0 & 0 & 0 & 1 \end{array}\\ \begin{array}{cccc} 1 & 0 & 0 & 0\\ 0 & 1 & 0 & 0\\ 0 & 0 & 1 & 0\\ 0 & 0 & 0 & 1 \end{array} & \begin{array}{cccc} 0 & 1 & 0 & 1\\ 1 & 0 & 1 & 0\\ 0 & 1 & 0 & 1\\ 1 & 0 & 1 & 0 \end{array} \end{array}\right)$$
where the first ($A^{\left(1\right)}$) and second ($A^{\left(2\right)}$) block matrices along the diagonal (i.e., the top-left corner and bottom-right corner) represent the adjacency matrices of the layer $L_{1}$ and $L_{2}$ respectively. The top-right and bottom left block matrices represent the connection between the layers. Using the supra-adjacency matrix in (Equation 19), one can define the sets of ${\left\{B_{x}\right\}}_{x=1}^{8}$ along with the set of functions ${\left\{f_{x}\left(r\right)\right\}}_{x=1}^{8}$ and then simulate the evolution of the QW on the multilayer network by updating the elements of the block matrix in (Equation 9).

### 5.1. Probability Distribution

We perform the UBCRW and the Fourier walk on the toy multilayer network structure and investigate the probability of finding the walker on each layer after 100 steps. The UBCRW is initialized from the vertex 1, and the Fourier walk is initialized from two localized initial states of the form ${|1\rangle }_{p}⊗{|3\rangle }_{c}$ and ${|1\rangle }_{p}⊗{|5\rangle }_{c}$. We place the quantum walker at the position state ${|1\rangle }_{p}$ (vertex 1), attaching the coin state of ${|3\rangle }_{c}$ and ${|5\rangle }_{c}$ to the walker. We select these localized initial states to replicate the conditions similar to those in the UBCRW, which enables a meaningful comparison between the unbiased CRW and QW on the toy multilayer model. After 100 steps, we calculate the probability of finding the walker at each layer by summing the probability of finding the walker at each node corresponding to a given layer. An interesting distinction between the unbiased classical and quantum walkers is illustrated in Figure 2. For the case of UBCRW, the probabilities of finding the walker on the top and bottom layers eventually stabilize to a steady state as time progresses (Figure 2a). Conversely, the Fourier walk displays dynamic changes in probability over time (Figure 2b,c).

It is important to note that, in the case of UBCRW, the probability of locating the walker on the top layer is consistently higher than that of the bottom layer, which is expected since the top layer comprises a complete graph. However, in the Fourier walk, certain time steps exist where the probability of locating the walker on the bottom layer surpasses that of the top layer. For instance, in Figure 2c, one can see certain time steps where the probability of finding the walker on the bottom layer is higher than that of the top layer. In addition, from Figure 2b,c, one can identify that the different initial coin states of the Fourier walk can control the temporary transition of the quantum walker from top layer to bottom layer. Such a behavior has no analogy to the UBCRW. The ability of the Fourier walker to temporarily transit from the top layer to the bottom, with higher probability, implies that the Fourier walk could explore a broader portion of the multilayer structure compared to the UBCRW. This enhanced exploration could be useful when searching through large, complex databases represented as multilayer networks. In Section 6, we further examine this behavior by employing different types of larger multilayer networks.

In Figure 2b,c, we have seen that the quantum probability is fluctuating. This behavior is attributed to the unitary characteristic of QWs, which prevents the quantum walker from reaching a steady state [30]. Hence, to obtain an idea of the static picture, one can calculate the time-averaged probability of finding the quantum walker at vertex *x* defined by
(20)$${\bar{P}}_{T}\left(x\right)=\frac{1}{T}\sum_{t=0}^{T-1}P_{q}\left(x,t\right)$$
where T∈N [3]. Note that when *T* becomes larger, ${\bar{P}}_{T}\left(x\right)$ becomes a better measure that depicts the static picture. In Figure 3, we illustrate the time-averaged probabilities for a time period of T=100 for Fourier and Grover walks on the toy network model. For the sake of comparison, we calculate the probability profile of the UBCRW as well (Figure 3a). The vertical axis of the heatmaps in Figure 3 represents the initial node from which the walker starts the walk, while the horizontal axis represents the target node where the walker ends the walk. For both Fourier and Grover walk, we consider two cases. In the first case, we initiate both Fourier and Grover walkers from the localized initial state of the form $|\phi_{1}{\rangle \equiv |x\rangle }_{p}⊗\frac{1}{\sqrt{d_{x}}}\sum_{r=1}^{d_{x}}{|f_{x}\left(r\right)\rangle }_{c}$ where x∈{1,…,8}. We start both QWs from each node by attaching a uniform superposition of coin states. Then, for each node, the time-averaged probability for a time period of T=100 is calculated using (Equation 20). The results are given in Figure 3b,c. In the second case, we repeat the same procedure by using the localized initial state of the form $|\phi_{2}{\rangle \equiv |x\rangle }_{p}⊗\left(\frac{i}{\sqrt{d_{x}}}|f_{x}{\left(1\right)\rangle }_{c}+\frac{1}{\sqrt{d_{x}}}\sum_{r=2}^{d_{x}-1}|f_{x}{\left(r\right)\rangle }_{c}-\frac{i}{\sqrt{d_{x}}}{|f_{x}\left(d_{x}\right)\rangle }_{c}\right)$ where x∈{1,…,8}. The results are given in Figure 3d,e. According to Figure 3a, the classical walker tends to stay on the top layer of the toy model after 100 time steps, irrespective of its initial node. Since the top layer comprises a complete graph, we can expect this result. The time-average probability profile of the Fourier walk also exhibits similar behavior to the classical walker when initiated from $|\phi_{1}\rangle$. That is, the time-average probability of finding the Fourier walker on the top layer is relatively higher than that of the bottom layer, irrespective of the initial node.

However, when the Fourier walker is initiated from $|\phi_{2}\rangle$, we can confine the walk to a particular layer with a higher probability, which is visible in Figure 3d. When initiated from $|\phi_{1}\rangle$, Grover walk exhibits no significant behavior, but for the initial condition of $|\phi_{2}\rangle$, Grover walker tends to stay, with a higher probability, at the initial position and the corresponding position on the other layer. As a result, one can see a sharp line along the diagonal of the grid in Figure 3e and two lines parallel to this sharp diagonal line. Hence, one can control the quantum dynamics on the multilayer network by changing the initial state, which has no direct analogy in UBCRW.

### 5.2. Return Probability

Another interesting question one could ask related to CRWs or QWs on a multilayer network is that of how long it would take for the walker to return to its initial position. This could be understood as the recurrence of the walk on the multilayer network. Recurrence in a CRW is characterized by the Pólya number [31], which can be written as
(21)$$P=1-\frac{1}{\sum_{t=1}^{\infty }P\left(x_{0},t\right)}$$
where $P(x_{0},t)$ is the probability of finding the walker at the initial node $x_{0}$ at time step *t*. For P=1, the walk is identified as *recurrent*. Otherwise, the walk is called *transient*. Moreover, the expression
(22)$$P=1-\Pi_{t=1}^{\infty }\left[1-P\left(x_{0},t\right)\right]$$
can also be used as a definition for the Pólya number as it provides the same criteria for the classification of the walk [32]. With the formula given in (Equation 22), the notion of Pólya number can be extended to the study of recurrence in QWs [32,33]. For practical purposes, one can calculate the partial Pólya number using either (Equation 21) or (Equation 22) for a finite number of time steps $t=T_{p}$ rather than extending t→∞. We calculate the partial Pólya number for UBCRW as well as Grover and Fourier walks on the toy multilayer network by choosing a set of finite time steps $T_{p}\in \left\{0,5,10,\ldots ,200\right\}$ with a gap of 5 units. Our purpose is to make an estimation of the convergence of the Pólya number for each walk. We initialize the walker from node 1, and for both Grover and Fourier walks, the initial coin state is chosen as the uniform superposition of coin states (i.e., ${|1\rangle }_{p}⊗\frac{1}{\sqrt{d_{1}}}\sum_{r=1}^{d_{1}}{|f_{1}\left(r\right)\rangle }_{c}$). Figure 4 shows the convergence of the partial Pólya number for UBCRW, Grover, and Fourier walks. According to Figure 4, Grover and Fourier walkers exhibit recurrence during the progression of the walk on the toy multilayer structure. However, the Grover walker returns to its initial position faster than the Fourier and unbiased classical walker.

### 5.3. Impact of Decoherence

This section is devoted to study the impact of decoherence on the quantum dynamics of the walker that propagates on the toy mulitlayer network model. With regard to this, we study the impact of decoherence that arises from randomly broken links. In the context of QWs on graphs, broken link decoherence specifically relates to the loss of coherence in a QW due to imperfections or disruptions in the graph structure [34,35]. This can occur when edges in the graph are altered or removed, creating discontinuities in the QW. These disruptions can be caused by various factors, including physical imperfections, noise, or intentional modifications to the graph.

We perform a Fourier walk for 100 time steps on the toy mulitlayer network model while breaking the connection between nodes 1 and 3 with a 0.5 probability at each time step. Afterward, the mean probability distribution was calculated by averaging over 1000 trials. Figure 5c displays the mean probability distribution of the Fourier walk over 100 time steps when subjected to the broken link decoherence model. Additionally, for comparison purposes, the probability distributions of the UBCRW and the standard Fourier walk are also presented in Figure 5a and Figure 5b, respectively. According to Figure 5a, the classical walker tends to stay on the top layer with higher probability after 100 time steps, a behavior which we have consistently seen in Figure 2 and Figure 3. On the other hand, the standard Fourier walk exhibits a very different probability profile compared to the UBCRW, as shown in Figure 5b.

However, when subjected to a broken link decoherence model, the classical signature emerges in the average probability distribution of the Fourier walk as depicted in Figure 5c. Recall that, in this study, we have realized the broken link decoherence model by breaking the connection between nodes 1 and 3 of the toy multilayer network model with a 0.5 probability at each time step. That is, we allow to alter only a single edge in the toy multilayer network. Yet, the effect of decoherence substantially impacts the dynamics of the walk, eventually converging it to the classical distribution. This implies that QWs on a multilayer network could be sensitive to decoherence models like broken links.

## 6. Numerical Implementation on Synthetic Multilayer Networks

We next apply our model to perform QWs on six different two-layered multiplex networks, each consisting of 100 nodes. The top and bottom layers of each multiplex network are constructed from the combinations of scale-free (SF), complete (CP), and star networks with 50 nodes. Moreover, the walker is initiated from a randomly chosen node for the CP and SF networks and from the hub node in the case of the star network. First, we perform the Fourier walk on the two-layered multiplex networks, which is initiated from the localized state of ${|1\rangle }_{p}⊗\frac{1}{\sqrt{d_{1}}}\sum_{r=1}^{d_{1}}{|f_{1}\left(r\right)\rangle }_{c}$. For each time step, up to 100 time steps, we calculate the probability of finding the walker on each layer by summing the probabilities of finding the walker at each node corresponding to that layer (Figure 6). For comparison purposes, we perform the UBCRW on the same two-layered multiplex network structures (Figure A4). The unbiased classical walker is initiated from node 1.

By comparing the results in Figure 6 and Figure A4, one can conclude that the probability of finding the Fourier and classical walkers on most of the multiplex networks follow a similar trend. However, for the cases of CP-CP, CP-STAR, and STAR-STAR, Fourier walk shows slight differences in its probability profiles in the first few time steps compared to the UBCRW (Figure 6d–f and Figure A4d–f). Nonetheless, as time elapses, the overall trend of the probability profiles of the Fourier walker becomes similar to that of the UBCRW.

The probability profiles of the Grover walk on the six different multiplex networks are given in Figure 7. The Grover walker is also initiated from the localized state of ${|1\rangle }_{p}⊗\frac{1}{\sqrt{d_{1}}}\sum_{r=1}^{d_{1}}{|f_{1}\left(r\right)\rangle }_{c}$ and for each time step up to 100 time steps, we calculate the probability of finding the walker on each layer by summing the probabilities of finding the walker at each node corresponding to that layer. On average, for the cases of SF-SF and SF-STAR, the Grover walker tends to be on both layers with an equal probability. This can be identified from Figure 7a,c. Grover walker on the CP-STAR multiplex network behaves in a very similar way to a classical walker (Figure 7e and Figure A4e). A periodic behavior of the probability of finding the Grover walker on top and bottom layers can be seen on CP-CP and STAR-STAR multiplex networks (Figure 7d,f). The periodicity reflects Grover walker’s coherent oscillations between the network layers and could be influenced by the topology and connectivity of the CP-CP and STAR-STAR multiplex network. Further research and analysis are essential to unlock the full potential of these insights for practical quantum applications.

In addition to exploring probability profiles, we have studied the recurrence probability of the Fourier, Grover, and unbiased classical walkers on the six different multiplex networks. We calculate the partial Pólya number for the UBCRW, Grover, and Fourier walks by choosing a set of finite time steps $T_{p}\in \left\{0,5,10,\ldots ,100\right\}$ with a gap of 5 units. Our purpose is to make an estimation of the convergence of the Pólya number for each walk. We initialize each walker from node 1, and for both Grover and Fourier walks, the initial coin state is chosen as the uniform superposition of coin states (i.e., ${|1\rangle }_{p}⊗\frac{1}{\sqrt{d_{1}}}\sum_{r=1}^{d_{1}}{|f_{1}\left(r\right)\rangle }_{c}$). Figure 8 shows the convergence of the partial Pólya number for the UBCRW, Grover, and Fourier walks. From Figure 8, one can identify that the Grover walk exhibits recurrence on most of the network structures. On the other hand, Fourier and unbiased classical walkers show no recurrence within 100 time steps. For all the plots in Figure 8, initially, the convergence of the partial Pólya number of the Fourier walk is low compared to that of the unbiased classical walker. However, as time elapses, the convergence of the Fourier walk surpasses that of the UBCRW, except for the case of the STAR-STAR multiplex network.

We apply the broken link decoherence model for the six different multiplex networks as well by following the same procedure described in Section 5. To realize the broken link decoherence model, we have removed some intralayer edges of the multiplex networks randomly and have calculated the average probability distribution of the QW after 100-time steps by averaging over 1000 trials. In Section 5, we observed the emergence of classical signature in the probability distribution even for a single broken link on the toy multilayer network. However, since the number of nodes in the six different multiplex networks is relatively large, we could not observe a fast convergence to the classical behavior when a single edge is broken. Nonetheless, when the number of broken links increases, the convergence to the classical distribution seems to be faster (Figure A3). This appears to imply that the impact of decoherence depends on the number of broken edges—a claim that should be further quantified and generalized in future research. While our investigation has shed some light on decoherence in multilayer networks, it is important to acknowledge that the scope of this research is not fully comprehensive. There remains significant potential for future studies to expand upon these findings to gain a deeper understanding of how decoherence impacts the propagation of the quantum walker on multilayer networks.

## 7. Discussion

This article studied the dynamics of discrete-time quantum walks on multilayer networks. We derived a recurrence formula for the coefficients of the wave function of a quantum walker on an undirected graph with finite nodes. Then, by extending this formula to include extra layers, we developed a simulation to mimic the evolution of the quantum walker on a multilayer network. While multilayer networks have been studied in the context of CTQWs, to the best of our knowledge, there is a lack of literature related to the case of DTQWs. Hence, the prime objective of this study was to present a comprehensive mathematical framework to model DTQWs on multilayer networks with the aim of bridging this gap. In this regard, we employed our mathematical model to analyze the time-averaged probability and the return probability of the quantum walker on multilayer networks in relation to the Fourier and Grover walks. Moreover, we studied the impact of decoherence on the progression of Fourier walk on multilayer networks. For the sake of clarity and readability, first, we used a toy multilayer network to conduct our analysis. Later, we extended our analysis to much larger synthetic multilayer networks. Our study revealed that the Grover walk on a multilayer network exhibits rich dynamics. For instance, the Grover walker displays a periodic behavior of occupying the top and bottom layers of a two-layered multiplex network constructed from a complete graph or a star graph. Further research and analysis are essential to unlock the full potential of these insights for practical quantum applications, e.g., for quantum computation and quantum communication. Moreover, in relation to the recurrence probability, the Grover walker returns to the initial position faster than both Fourier and unbiased classical walkers. In the context of general QWs, the recurrence probability has a deep link to the localization property, playing a pivotal role in diverse applications, including quantum search algorithms and topological insulators [36,37,38,39]. Hence, there seems to be significant potential for future studies related to the return probability on multilayer networks.

Another finding of this study is that the QWs on multilayer networks are vulnerable to decoherence arising from randomly broken links. Multilayer networks with a smaller number of nodes are sensitive to defects, even on a single edge. However, for larger multilayer networks, the tolerance of the QWs for decoherence appears to decrease as the number of broken links increases. Nonetheless, this observation should be further quantified and generalized in future research.

To experimentally realize quantum walks on multilayer networks, we need to implement a position-dependent coin. This can be archieved, e.g., by employing the method recently reported in [40]. In this experiment, a cascaded interferometric network was constructed incorporating birefringent calcite beam displacers to perform the conditional position shifts. Position-dependent coin operations were realized using various wave plates placed in the specific spatial modes of the heralded single photons traversing the setup.

As a future extension of this study, one could explore how other forms of decoherence models, such as Pauli channels, as well as amplitude and phase damping in the coin degree of freedom, impact the QWs on multilayer networks. In summary, we anticipate that the mathematical analysis we have performed here may have a profound influence on a broad spectrum of problems that can be modeled or assisted by DTQWs on multilayer networks.

## Figures and Tables

**Figure 1 entropy-25-01610-f001:**
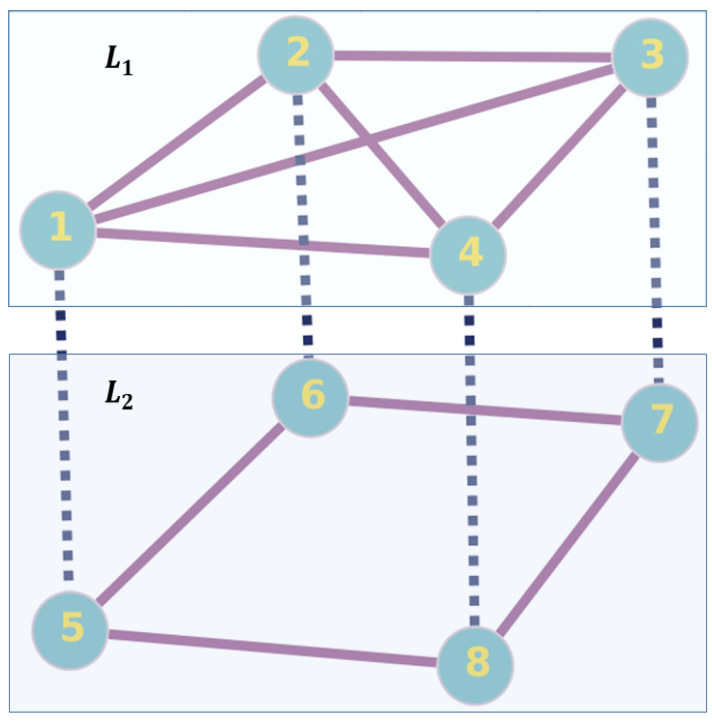
A multilayer network with four vertices and two layers named $L_{1}$ and $L_{2}$. The set of labels {x,x+4} where x∈{1,2,3,4} represent the same entity in different layers. The solid lines connect the vertices within each layer, and the dotted lines connect interlayer. The top layer represents an undirected regular graph, and the bottom layer represents an undirected connected graph. This schematic diagram is inspired by [6,7].

**Figure 2 entropy-25-01610-f002:**
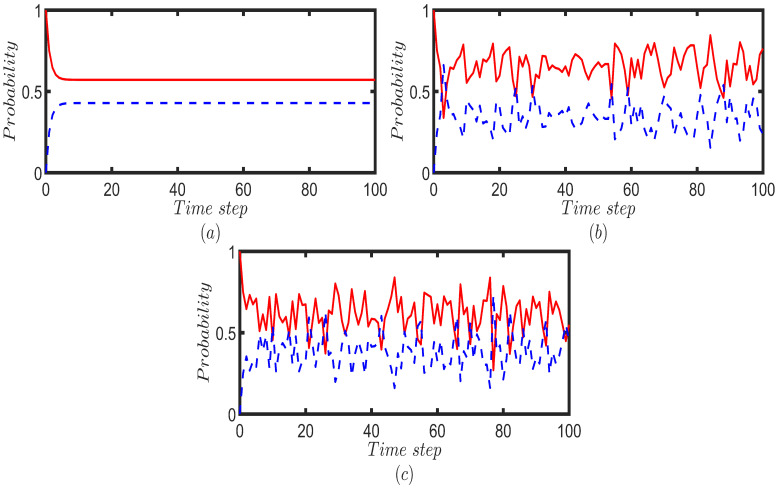
Probability of finding the walker on the top layer (red solid line) and bottom layer (blue dotted line) for each time step up to 100 steps (**a**) unbiased CRW is initiated from vertex 1. Fourier walk is initiated from (**b**) ${|1\rangle }_{p}⊗{|3\rangle }_{c}$ and (**c**) ${|1\rangle }_{p}⊗{|5\rangle }_{c}$.

**Figure 3 entropy-25-01610-f003:**
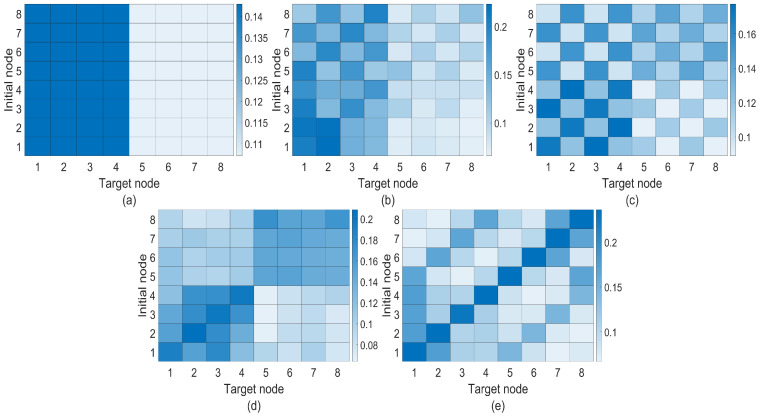
Heatmaps are depicted for (**a**) unbiased CRW, (**b**,**d**) Fourier walk, and (**c**,**e**) Grover walk. The vertical axis shows the initial node from which the walker starts the walk, and the horizontal axis shows the target node where the walker ends the walk. Each square corresponding to the Fourier and Grover walks indicates the value of time-averaged probabilities for a time period of T=100. For the unbiased CRW, each square corresponds to the probability after 100 time steps. Both Fourier and Grover walks in (**b**,**c**) are initiated from the localized position state $|\phi_{1}{\rangle \equiv |x\rangle }_{p}⊗\frac{1}{\sqrt{d_{x}}}\sum_{r=1}^{d_{x}}{|f_{x}\left(r\right)\rangle }_{c}$ and for (**d**,**e**) walks are initiated from the localized position state $|\phi_{2}{\rangle \equiv |x\rangle }_{p}⊗\left(\frac{i}{\sqrt{d_{x}}}|f_{x}{\left(1\right)\rangle }_{c}+\frac{1}{\sqrt{d_{x}}}\sum_{r=2}^{d_{x}-1}|f_{x}{\left(r\right)\rangle }_{c}-\frac{i}{\sqrt{d_{x}}}{|f_{x}\left(d_{x}\right)\rangle }_{c}\right)$. Note that, $|\phi_{1}\rangle$ and $|\phi_{2}\rangle$ have a uniform superposition of coin states, yet $|\phi_{2}\rangle$ contains some complex coefficients.

**Figure 4 entropy-25-01610-f004:**
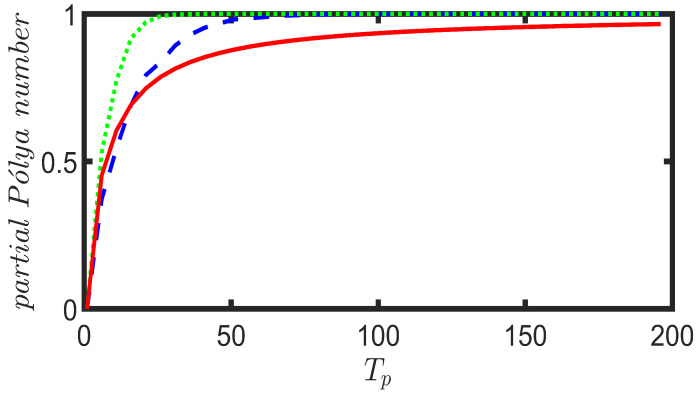
Convergence of the partial Pólya number for Grover (green dotted line), Fourier (blue dash line), and unbiased classical (red solid line) walkers on the toy multilayer network. The Partial Pólya number is calculated by choosing a set of finite time steps $T_{p}\in \left\{0,5,10,\ldots ,200\right\}$ with a gap of 5 units. Each walk is initiated from node 1, and for both the Grover and Fourier walks, the initial coin state is chosen as the uniform superposition of coin states.

**Figure 5 entropy-25-01610-f005:**
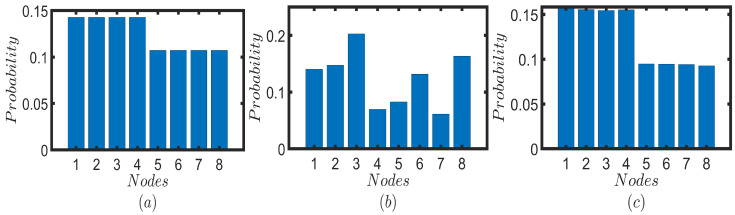
Probability distributions of the (**a**) unbiased CRW and (**b**) the standard Fourier walk after 100 time steps on the toy multilayer network. The unbiased CRW is initiated from node 1, and the Fourier walk is initiated from the localized state of ${|1\rangle }_{p}⊗{|2\rangle }_{c}$. (**c**) The mean probability distribution of the Fourier walk subjected to the broken link decoherence model is generated by breaking the connection between nodes 1 and 3 of the toy multilayer network with a 0.5 probability at each time step and by averaging over 1000 trials.

**Figure 6 entropy-25-01610-f006:**
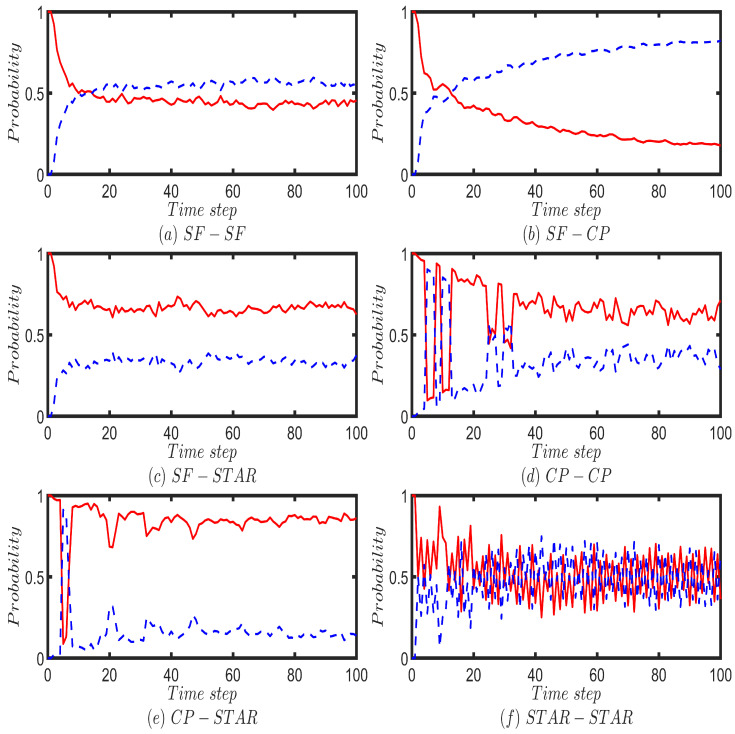
Illustrates the probability of finding the Fourier walker on the top layer (red solid line) and the bottom layer (blue dotted line) of six different two-layered multiplex networks, each consisting of 100 nodes. The top and bottom layers of each multiplex network are constructed from combinations of scale-free (SF), complete (CP), and star networks with 50 nodes. For the case of SF-SF, two different scale-free networks are chosen. Fourier walk is initiated from the localized position state of the form ${|1\rangle }_{p}⊗\frac{1}{\sqrt{d_{1}}}\sum_{r=1}^{d_{1}}{|f_{1}\left(r\right)\rangle }_{c}$ and for each time step up to 100 steps, the probability of finding the walker on a given layer is calculated by summing the probabilities of finding the walker at each node corresponding to that layer.

**Figure 7 entropy-25-01610-f007:**
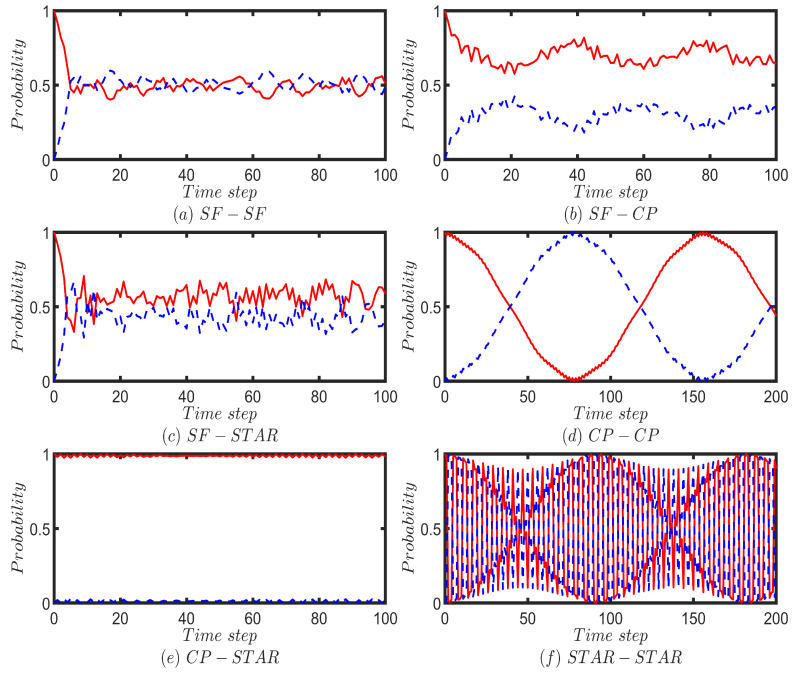
Illustrates the probability of finding the Grover walker on the top layer (red solid line) and the bottom layer (blue dotted line) of six different two-layered multiplex networks, each consisting of 100 nodes. The top and bottom layers of each multiplex network are constructed from combinations of scale-free (SF), complete (CP) and star networks with 50 nodes. For the case of SF-SF, two different scale-free networks are chosen. Grover walk is initiated from the localized position state of the form ${|1\rangle }_{p}⊗\frac{1}{\sqrt{d_{1}}}\sum_{r=1}^{d_{1}}{|f_{1}\left(r\right)\rangle }_{c}$ and for each time step up to 100 steps, the probability of finding the walker on a given layer is calculated by summing the probabilities of finding the walker at each node corresponding to that layer. In cases (**d**,**f**), the time step has been extended to 200 steps to improve the clarity and visibility of the plot shapes.

**Figure 8 entropy-25-01610-f008:**
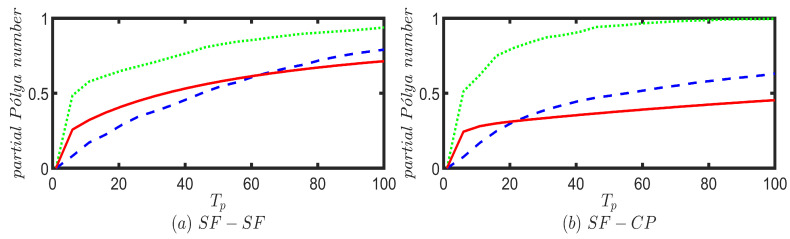
Convergence of the partial Pólya number for Grover (green dotted line), Fourier (blue dash line), and unbiased classical (red solid line) walkers on six different two-layered multiplex networks, each consisting of 100 nodes. The top and bottom layers of each multiplex network are constructed from combinations of scale-free (SF), complete (CP), and star networks with 50 nodes. For the case of SF-SF, two different scale-free networks are chosen. Partial Pólya number is calculated by choosing a set of finite time steps $T_{p}\in \left\{0,5,10,\ldots ,100\right\}$ with a gap of 5 units. Each walk is initiated from node 1, and for both Grover and Fourier walks, the initial coin state is chosen as the uniform superposition of coin states.

## Data Availability

The data and code can be made available upon request.

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
