# Peer review of "Discrete-Time Quantum Walk on Multilayer Networks"

_entropy, 2023, doi:10.3390/e25121610_

Round 1

Reviewer 1 Report

Comments and Suggestions for Authors

Please check the attached pdf.

Comments on the Quality of English Language

Good. Minor editing is required.

Reviewer 2 Report

Comments and Suggestions for Authors

Multilayer network is known to be a potent platform which does pave a way in order to study the interactions among entities in various networks with multiple types of relationships. The authors have studied the dynamics of  discrete-time quantum walk on a multilayer network in a detailed manner. I find it to be of particular interest that the authors presented an analysis of the impact of decoherence on the quantum transport, which sheds light on how environmental interactions may impact the behavior of quantum walkers on multilayer network structures.

Generally speaking, the paper is written sufficiently satisfactorily and the citations are reasonable adequate. I, therefore, recommend its publication, possibly some careful and thorough double-checking, text-editing and proofreading of the entire paper.

Author Response

We thank the Reviewer for the supportive comments and positive assessment. As recommended by the Reviewer, we have made a careful and thorough double-checking, text-editing and proofreading of the entire paper. To enhance the readability of the paper we made the following minor changes to the revised manuscript.

Change 1: The word “nodes” is replaced by the word “edges” in line 21

Change 2: The phrase “the important models of ” is added to line 27

Change 3: We removed the following two sentences from the “Introduction” section 

“The most general form of networks and multilayer networks are mathematically represented using the notion of a graph [5]. Hence, its worth exploring the research work on DTQWs that incorporate graphs.”

Change 4: The word “graph” is replaced by the word “network” in lines 31 and 32

Change 5: The phrase “Extending the idea given in” is replaced by the phrase “Redefining the action of the conditional swap operator” in line 36.

Change 6: We removed the equation (3) and embedded that information to the main text by adding the sentence, “For each position x, the coefficient αx,r(t)=0 whenever r∉ Bx.” After that we renamed the equations accordingly and corrected some additional minor typos.

Round 2

Reviewer 1 Report

Comments and Suggestions for Authors

Please check the attached pdf. 

Comments on the Quality of English Language

Minor editing is required. 
